# Effects of Mobile Identity on Smartphone Symbolic Use: An Attachment Theory Perspective

**DOI:** 10.3390/ijerph192114036

**Published:** 2022-10-28

**Authors:** Jie Lou, Nianlong Han, Dong Wang, Xi Pei

**Affiliations:** 1Department of International Business, School of Foreign Languages and Business, Shenzhen Polytechnic, Av. Liuxian, Shenzhen 518055, China; 2School of Geography and Tourism, Huizhou University, Huizhou 516007, China

**Keywords:** smartphone, mobile technology, symbolic use, attachment theory, identity, mental health, self-concordance model

## Abstract

Smartphones are not only multifunctional tools but also users’ personal extensions and identity symbols, as they are constantly with users and highly visible to the public while in use. Due to this public property as well as the close bond between smartphones and users, they are frequently used for personal identity expression besides functional purposes. The current study conceptualizes such behavior as symbolic use and aims to understand it. Anchoring on the attachment theory, mobile identity is postulated as an important antecedent of symbolic use. Mobile identity in turn is formed by mobile symbolism and mobile design esthetics. The research model was tested by a hybrid of both online and offline survey with 271 valid responses. SEM analysis was used to test the research model and SPSS was used for descriptive statistics. The results confirmed the role of mobile identity in affecting smartphone symbolic use. Additionally, individual materialism was confirmed as a moderator using hierarchical analysis. By defining and explaining smartphone symbolic use, this study clarifies the unique characteristics of the smartphone usage context as compared to non-portable technologies, thereby enriching the mobile usage literature and the application of attachment theory. It also defines the boundary condition of attachment formation by studying the contingent role of individual characteristics.

## 1. Introduction

Smartphones have permeated modern life and users rely on them for everything ranging from socializing to entertaining. As a result, users develop strong attachment to them and use them extensively [1]. According to the latest report of CNNIC (China Internet Network Information Center), 99.7% of netizens in China use smartphones to access the internet, with an average screen time of 28.5 h per week [2].

Smartphones distinguish themselves from other non-portable personal technologies (e.g., desktop, printer, TV, etc.) in two aspects: (1) they are with users almost all the time through various daily events [3,4,5], leading to heavy reliance and a close bond [6,7]. However, such connection hardly exists between users and non-portable technologies. (2) As smartphones are often used in the public and have high visibility in use, they contain immense symbolic meanings to users and become important channels for users to express their personal identity [8,9]. Therefore, smartphones are regarded as personal extensions with symbolic meanings [8,9] and often used for social outcomes and identity expression [10,11] besides functional purposes. For example, users personalize smartphones with various phone cases, leading to a rapidly growing phone case industry worth an estimated USD 21.61 billion [12].

In examining smartphone usage, previous studies have been mainly concerned with the functional aspect, i.e., using various functions afforded by the flexible operating system and diverse applications [1]. Relevant variables include screen time, usage intensity, frequency, etc. [13,14,15]. Realizing the special features of smartphones as personal extension, scholars have proposed a differentiate usage of mobile technology for identity purposes compared to the usage for pragmatic purposes [16]. However, the non-functional aspect of smartphone usage and its working mechanism remains understudied. Given the pervasiveness and uniqueness of such behavior, plus the special role of smartphones as signifiers of identity, it is pertinent to bridge this gap. This paper thus strives to answer three research questions: (1) How can we conceptualize and measure smartphone non-functional usage? (2) How is such behavior formed? (3) Is there any contingent factor that defines the boundary conditions of such behavior?

To address the first question, smartphone symbolic use was conceptualized to reflect non-functional smartphone usage and it can be captured by three elements, i.e., mobile personalization, public display, and protective behavior. In answering the second question, the attachment theory was drawn upon as the theoretical framework to explain how mobile identity and its antecedents influence smartphone symbolic use. To answer the third question, we integrated the self-concordance model with attachment theory to explore the influence of user characteristics (individual materialism) in the process.

## 2. Literature Review

### 2.1. Symbolic Use of Smartphones

While functional usage of technology has been a regular topic in previous studies, non-functional interaction with IT artifacts is a relatively new one. However, there are studies that have already looked into the phenomenon. For example, in organizational technology studies, system usage has been revealed to serve not only functional purposes, but also symbolic ones by gaining intangible benefits for companies, such as reputation, legitimacy, positive impression, etc. [17,18,19]. In the realm of mobile technology research, especially studies adopting uses and gratification theories, scholars have proposed to differentiate usage of mobile technology for identity purposes from the usage for pragmatic purposes [16]. It was noted that users treat mobiles as “signifiers of identity” to help convey their actual or ideal image due to their nature as highly personal belongings [20]. Furthermore, fashion and status have been identified as crucial motives for mobile technology usage [9,20,21]. However, such a phenomenon has not been systematically studied, and the drivers behind the non-functional usage behavior are unknown.

Given the significant symbolic value of smartphones due to their personal and public nature, we conceptualize “symbolic use” to reflect users’ employment of smartphones for non-functional purposes and social outcomes, which are mostly related to self-identity expression. In searching for the specific content underlying symbolic use, three types of typical behaviors were identified.

Smartphones are often viewed as part of users’ extended self [21]. It is noted in the literature that users treat mobile devices as tools to communicate their personality and as a fashion statement [10]; thus, they like to customize the appearance of phones [16,22]. Such customization includes both interior decoration such as personalizing ringtones and wallpapers [16,22] and exterior decoration of physical appearance [10]. Due to smartphones’ high visibility in use, such behavior is especially common and important to users compared with non-portable technologies. **Mobile personalization** is thus included as the first dimension of symbolic usage. It is defined as the extent to which users customize smartphone appearance to express themselves through both interior and exterior decoration [10].

Dedeoglu [20] found that for users who regard mobile technologies as symbols of popularity, prestige, and personality expression, they tend to display their smartphones more often in public to signify their actual or ideal image. Such behavior is also common with other material possessions that contain symbolic meaning [11], since enhancing social visibility can facilitate identity expression and impression management [23]. **Public display** is thus included as the second dimension of symbolic use. It is defined as the extent to which a user likes to display smartphones in public, even with no specific intention of using their functions.

Finally, individuals are naturally inclined to protect and nurture their self-identity; therefore, they exhibit protective behavior towards the material possessions that are regarded as their personal extensions [24]. Specifically, such behavior is exemplified by greater efforts in maintaining the possession and emotional difficulty in accepting the deterioration or loss of the object [25]. Putting this in the mobile context, as smartphones are users’ important personal extensions and users tend to protect their personal identity, they handle smartphones that can facilitate their identity expression with great care. **Protective behavior** is thus included as the third dimension of the symbolic usage construct. Please refer to Table 1 for summary.

### 2.2. Smartphones and Attachment Theory

Classic technology usage theories such as TAM (technology acceptance model), the value-based adoption model of technology [27,28,29], and the IS (information system) continuance model [30,31] have been widely adopted to explain mobile phone functional usage. However, we argue that as smartphones distinguish themselves as hybrids of both instrumental tools and symbolic items [8,9], users have distinct decision processes. Especially when we examine non-functional usage, the rationale behind the symbolic usage behavior is clearly different from previous task-oriented behavior. Therefore, this paper takes into consideration the close bond between smartphones and users, and proposes that a relationship-based perspective, i.e., attachment theory is more appropriate in explaining smartphone symbolic use.

The attachment theory, which was originally developed in the parent–infant relationship area, offers a “need satisfaction-attachment formation-resource allocation” framework. Specifically, human beings’ innate attachment behavioral system keeps them close to attachment targets (e.g., caregivers) who respond to their needs, as a means of seeking protection and attaining emotional safety [32]. In the need fulfilment process, the attachment targets become highly trusted and irreplaceable, leading to the formation of attachment [33]. As a result, an intense and committed relationship is formed, which individuals will strive to maintain by allocating emotional, cognitive, and behavioral resources towards the attachment target [34,35].

Besides attachment to parents and lovers [36], attachment theory is also widely applied in marketing research to study peoples’ attachment to material possessions [22,37,38], brands [37,38,39,40], recreational places [41,42,43], etc. The notion of attachment is also not new to mobile studies. As mobile phones are barely separated from users, deep relationships are cultivated [7] and users regard them as personal extensions of themselves, leading to strong emotional attachment [22,44].

Following the “need satisfaction-attachment formation-resource allocation” framework of attachment theory, the marketing literature suggests that users’ attachment to products, particularly interactive items, can be cultivated if three needs are fulfilled; i.e., the needs of enriching, gratifying, and enabling the self [40,42,45]. Enriching the self requires the product to represent and express users’ self-identity to receive social approval; gratifying the self requires offering esthetic joy and sensory pleasure; enabling the self requires supporting users’ goal achievement and providing a sense of self-efficacy [46]. As the dependent variable of this study is symbolic use, which concerns non-functional interaction with smartphones, the needs of enriching and gratifying the self are considered more relevant as they deal with identity expression and hedonic enjoyment. However, enabling need is regarded as irrelevant as it is mainly related to functional usage. We thus examined **enriching and gratifying factors** as the main antecedents of mobile attachment in this paper.

There are several different conceptualizations of the attachment construct. In marketing and psychology, it is usually treated as a one-dimensional construct, referring to the emotional bond between users’ self-concept and products [45,47,48]. Such a bond involves complex emotional feelings such as happiness from self–target proximity, pride from target–self display, and anxiety in self–target separation [37]. However, such conceptualization is quite subtle as it is not specific about the way users and attachment targets are actually connected.

Studies of attachment to recreational places address this concern by conceptualizing the attachment constructs as two-dimensional, i.e., concerning place dependence and place identity [44,46,49,50]. Place dependence refers to users’ functional reliance on the attachment target, and it emphasizes the importance of the target as an instrumental tool in goal achievement [42]. However, the strong bond between user and attachment target cannot be fully reflected by the functional connection alone [51]. It is supplemented by the second dimension, place identity, which refers to one’s self-identity that is defined by the place one visits [45]. Place attachment studies note that besides achieving functional and experiential goals, people visit recreational places because they identify with a place in that such a place creates and maintains visitors’ self-identity. Such connection is emotional and symbolic in its nature [52].

Despite great differences, what smartphones and recreational places have in common is that they not only help users’ achieve pragmatic goals, but also convey important symbolic meaning to users/visitors. This two-dimensional conceptualization of attachment was also applied to the mobile context and the authors name the dimensions as “mobile functional dependence” and “mobile identity”, respectively [1]. While the former concerns the extent to which users rely on smartphones to accomplish functional goals, the latter reflects the extent to which individuals regard smartphones as a part of their self-identity, which communicates to others who they are and gives meaning to their lives. Again, as this paper mainly studies non-functional-related symbolic use, “mobile functional dependence” is regarded as less relevant. We thus focus on “mobile identity” to reflect the symbolic and emotional aspect of users’ attachment to their smartphones.

### 2.3. Self-Concordance Model

Attachment research has elaborated on the attachment formation process through the mechanism of need satisfaction [40,42,45]. However, it is unknown whether different needs are equally important for different users. It is imperative to study such contingent effects because understanding which type of need is more important to certain users can shed light on how to design and market smartphones to appeal to distinct user segments.

Towards this end, we draw from the self-concordance model; a model about goal striving and need satisfaction [47]. The concept of goal self-concordance plays a central role in the model. It is defined as the extent to which the goals of attainment reflect users’ lasting interests and values [47]. Self-concordant goals are the ones that are pursued because of users’ intrinsic motivation or interest. Such goals have an internal locus of causality, i.e., they come from users’ self-choice instead of external pressure and can facilitate the need satisfaction process as well as benefit mental health [48]. In contrast, goals that are not consistent with individual value will bring users a certain degree of control [49], resulting in a compromise between the goal attachment outcome and mental health. The self-concordance theory thus suggests a moderating role (The moderating effect refers to a situation in which a moderator affects the direction or the strength of the relationship between an independent variable (IV) and a dependent variable (DV).) of goal self-concordance in goal outcome attainment (Figure 1). 

Putting this model in the mobile context, the various needs smartphones meet can be seen as the goals attained by smartphone users, and the attachment formation can be seen as the outcome of need satisfying experience. Yet whether the attained goal can lead to a positive need satisfying experience and benefit mental health depends on a moderator: goal self-concordance, which can be reflected by user characteristics [50].

As smartphones are becoming ubiquitous in daily life, their influence on user behavior is a hot topic in the area of public health. Both positive and negative consequences of smartphone usage on mental health have been discussed. On the positive side, using smartphones for workouts and food monitoring can benefit mental and physical health [53]. Meanwhile, easy access to social communication is positively related with psychological well-being [54,55]. However, there is also considerable research about the negative effect of smartphones usage on mental health, and causing anxiety, distraction, and depression, typically by overuse [54,56,57]. According to the self-concordance model, the different effects of using smartphones can be attributed to the level of goal–self concordance; i.e., when goals in using smartphones are consistent with users’ individual characteristics, a positive effect on mental health can be expected and vice versa.

## 3. Hypothesis Development

### 3.1. The Effect of the Enriching Factors

As mentioned in Section 2.2, enriching factors that enable smartphones to define, represent, and express users’ self-concept [40,42,45] are crucial for attachment formation. There are three premises for a product to express users’ self-concept: visibility in use, variability in use, and personalizability [58]. Smartphones are highly visible as they are frequently displayed and used in public spaces. Smartphones can be flexibly personalized both in terms of appearance and functions based on individual needs. To sum up, smartphones can serve as efficient tools of self-concept expression.

Self-concept is people’s idea of who they are; it includes two parts, the actual self and ideal self [39,59]. While the former reflects users’ perception of themselves in reality, the latter refers to people’s aspired images, which is related to their goals. Expressing users’ actual self is a self-verification process; i.e., allowing users to confirm and maintain their actual self to achieve actual self-congruence [39]. Consistent with existing studies, the attribute of smartphones to communicate actual self-image is called “self-expressive symbolism” [60,61]. Expressing users’ ideal self is a self-enhancement process, allowing users to show their ideal social image (e.g., cool/trendy/rich/high-tech) [61] to achieve ideal self-congruence [39]. The property of smartphones enabling ideal self-expression is called “categorical symbolism” [60] (Figure 2).

According to the place attachment literature, place identity is formed when a certain place supports visitors in defining and maintaining self-identity [62]. Similarly, it is argued that the formation of mobile identity, the emotional and symbolic aspect of mobile attachment, is facilitated when users take smartphones as a part of their identity that shows who they are and gives meaning to their lives.

When users perceive the self-expressive symbolism or categorical symbolism of smartphones is high, it means their smartphones are consistent with their actual self or ideal self. Under such circumstances, the user is more likely to regard the smartphone as a component of his/her self-identity [39] and thereby develop mobile identity. For example, a technology lover would perceive a cutting-edge smartphone as consistent with his/her self-identity, and thus be more inclined to take it as a component of his/her self-identity (mobile identity) compared with an outdated phone. Therefore, we propose H1a and H1b as follows:

**H1a:** 
*Self-expressive symbolism is positively related to mobile identity.*


**H1b:** 
*Categorical symbolism is positively related to mobile identity.*


### 3.2. The Effect of the Gratifying Factor

To gratify the self, products should offer sensory and esthetic pleasure [40,42,45]. Such pleasure largely derives from design esthetics or the visual appeal of smartphones [27,63]. This is especially important for smartphones compared with non-portable technologies as smartphones are always with the users and used in public, thus often being regarded as personal extensions to display self-identity [8,9]. However, such factors are largely absent in the mobile usage literature with few exceptions. It is noted that beyond functional aspects, users seek beauty and enjoyment in the process of interacting with technologies [64]. In this vein, human–computer interaction (HCI) research has extensively studied the influence of IT artifacts’ esthetic qualities on users’ evaluation of technology [27,61,65,66,67]. We thus take design esthetics as a gratifying factor in the research. Following previous research, both software design and hardware design are included in this construct [63,68].

Design esthetics influence attachment formation through the mechanism of symbolic association, which means the product design contains socially determined symbolic information about users [60]. Especially in the mobile context, if users think their smartphones are nicely designed, they tend to think the smartphone contains certain symbolic meaning (both expressive and categorical symbolism) that reflects their self-image and can function as a surrogate of identity expression (actual self or ideal self). This in term leads to the symbolic bond between users and smartphones (mobile identity). In a word, the perceived symbolism of the smartphone mediates (The mediation effect refers to a causal chain in which an independent variable (IV) affects a dependent variable (DV) via an intervening variable, i.e., the meditator.) the positive relationship between design esthetics and mobile identity. Therefore, H2 is formulated as follows:

**H2:** 
*Perceived design esthetics are positively related to mobile identity via the mediation of self-expressive symbolism and categorical symbolism.*


### 3.3. The Effect of Mobile Identity

Attachment evokes users’ emotional willingness to invest resources in the attachment target [40]. In the marketing area, this argument is evidenced by spending personal resources such as time, money, and effort to stay loyal to certain brands/products, and even recommending them to others [40,69]. Adapting this effect to the mobile context, a strong attachment to smartphones leads to users’ readiness to spend their physical and mental resources on their mobile devices.

Mobile identity, as the emotional and symbolic aspect of mobile attachment, reflects the degree to which users regard smartphones as a component of their self-identity. Based on attachment theory, individuals have the natural tendency to protect and nurture their self-identity [25]; they are ready to actively spend emotional, cognitive, and behavioral resources on the attachment target [37].

As shown in the previous attachment literature, a strong symbolic bond leads people to keep adjacent to the attachment target, protect it, and engage in non-functional customization frequently to showcase their personal identity [10,32,67]. Therefore, in the smartphone usage context, it is argued that the behavior to cultivate self-identity can be reflected by the three types of symbolic usage behavior: demonstrating public display, personalizing smartphones, and protecting smartphones.

Therefore, H3 is formulated as follows:

**H3:** 
*Mobile identity is positively related to smartphone symbolic use.*


### 3.4. The Moderating Effect of Individual Materialism

Given the special characteristic of smartphones as individual material possessions, one relevant individual characteristic to study is individual materialism. It reflects users’ opinion with regards to the role of material possessions in their lives [70]. Since individual materialism influences how people perceive the importance of symbolic meaning of possessions [71], it is argued that it might influence the effect of mobile symbolism, i.e., the enriching need. Individuals who rate highly on individual materialism place material possessions at the center of their lives based on their value system. Meanwhile, they see possessions as a means of achieving happiness and regard possessions as indicators of their own and others’ success [71]. Therefore, users with high individual materialism highly value the importance of products’ symbolic meaning, as material possessions are expressions of their actual identity and also channels to indicate their ideal identity [11].

Smartphones that carry self-expressive symbolism and categorical symbolism can help a user fulfill their needs of expressing their actual or ideal identity. Such needs fit well with the value system of those users with high individual materialism as they assign high symbolic importance to material possessions [71]. As a result, mobile identity is more likely to be cultivated. Therefore, H4a and H4b are formulated as follows:

**H4a:** 
*Individual materialism positively moderates the effect of self-expressive symbolism on mobile identity.*


**H4b:** 
*Individual materialism positively moderates the effect of categorical symbolism on mobile identity.*


The complete research model is presented in Figure 3.

## 4. Materials and Methods

### 4.1. Data Collection

A survey methodology was used in this study. It was suitable for this study because the variables in the study are mainly individual perceptions, self-reported use, and individual characteristics. It is hard to manipulate these variables in experiments and more suitable to measure them by surveys. A mixture of online survey and paper-based questionnaire was adopted.

The online channel used a survey website (https://www.wjx.cn/, accessed on 26 March 2020) to conduct snowball sampling via the referral of SNSs (social networking sites) in mainland China in April 2020. It lasted for two weeks and received 182 responses, of which 147 were valid. The offline channel was conducted in a Chinese university where students were recruited to fill in questionnaires in classrooms; 124 valid responses were received out of 136 total responses. Participants of both channels received RMB 20 (about USD 3) if their responses were valid. The criteria of filtering invalid response were as follows: (1) if the value of reverse-coded items was similar to other items of the same variable; (2) if there were more than 20 repeated answers; (3) if IP addresses were the same. The data collection process finally led to 271 valid responses out of 318 total responses (85.2%). As shown in Table 2, 52.1% were females (N = 141), and the most used smartphones were Apple (N = 84, 31.0%), Huawei (N = 58, 21.4%), and Xiaomi (N = 27, 10%). It was suggested a sample size of more than five times the number of indictors was valid for SEM model testing [72]. To make it more rigid, ten sample points per indicator were regarded as a widely adopted rule of thumb [73,74]. Given that there were 27 indicators of 8 latent variables in the SEM model, the sample size of 271 is regarded adequate.

### 4.2. Instrument

We adopted most measurement items from previous studies, with reasonable adjustment to better fit the smartphone context (Table 3). The items were rated following a seven-point Likert scale ranging from 1 = strongly disagree, to 7 = strongly agree. The items of “self-expressive symbolism” and “categorical symbolism” were adapted from [39] to access the extent to which smartphones reflected users’ actual or ideal self-identity, respectively. “Perceived design esthetics” were modeled as a second-order formative construct that included perceived hardware design esthetics and software esthetics. The items of “perceived hardware design esthetics” and “software design esthetics” were adapted from [63,65] to examine whether the user thought smartphone hardware and software design esthetically appealing. The items of “mobile identity” were adopted from [62] to examine if smartphones were taken as a component of users’ self-identity.

The dependent variable, smartphone symbolic use, was measured by three types of different behavior: public display, mobile personalization, and protective behavior. As the three behaviors do not have the same content, i.e., they are not interchangeable and do not co-vary with each other, symbolic usage was modeled as a second-order formative construct consisting of three first-order constructs. For “public display”, as there is no existing measure for this construct, four items were newly developed based on the construct definition. These four items emphasize whether a user likes to display their mobile device in public so that others can see it. The measurement for “mobile personalization” was developed based on the description of [10,16] to measure whether a user likes to customizes mobile appearance to express themselves. The final dimension, protective behavior, was adapted from [24,25]. Finally, the measurement items of the moderator “individual materialism” were adapted from [75]. Besides “perceived design esthetics” and “smartphone symbolic use”, which were modeled as second-order constructs, other constructs are all reflective.

As the symbolic usage of mobile technology involves non-functional interaction with mobile technology, the relationship between user and brand also influences such non-functional usage of a product [37]. Brand attachment, which describes whether there exists a close bond between users and certain products [37,39], is thus included as a control variable for smartphone symbolic use.

Moreover, some of users’ symbolic usage, such as protective behavior and personalization, is likely to be time sensitive; i.e., such behavior will decrease as the time of ownership becomes longer. Therefore, to control for the time effect of attachment formation, time since ownership was included as a control variable for smartphone symbolic use.

Four Ph.D. students and two non-experts were invited to conduct two rounds of card sorting to ensure content validity of the survey [76]. As the survey was conducted in mainland China, it was translated to Chinese followed by a pre-test to receive feedback about the wording, logic, translation of instruction, and questionnaire. Adjustments were made based on the comments of 20 participants.

## 5. Results

SEM (structural equation modeling) analysis was used to test the research model and hypotheses. SEM, as a second generation statistical analysis method, allows the examination of both (1) the relationships between observable variables and latent variables (measurement model), and (2) the relationships among latent variables (structural model) [77]. Specifically, SmartPLS 3.0 was used as the major software package, with SPSS (Software Package for Social Science, SPSS Inc. Released 2008. SPSS Statistics for Windows, Version 17.0. SPSS Inc., Chicago, IL, USA) 17.0 used for descriptive statistics.

Anderson and Gerbing [78]’s two-step approach was followed in analyzing data. In the first step, the measurement model was accessed to test the appropriateness of the instruments. In the second stage, the structural model was tested to see whether the hypotheses were supported. The moderating effect was also tested using PLS. A hierarchical process was established and two models (one with and one without interaction constructs) were compared.

### 5.1. Measurement Model

Validity of the measurement model was accessed by a combination of Cronbach’s alpha, composite reliability (CR), and average variance extracted (AVE) [79] (please refer to Table 4 for details). The indices easily reached the suggested threshold values of 0.70, 0.70, and 0.50, respectively [73].

When a construct’s square root of AVE is higher than its correlations with other constructs, discriminant validity is satisfactory [80]. As shown in Table 5, good discriminant validity of all constructs was guaranteed.

Convergent and discriminant validity were further assessed by the loadings and cross-loadings matrix. If the items load high on their corresponding constructs, good convergent validity is guaranteed and if there is great difference between the item loading on corresponding constructs and that of other constructs, good discriminant validity is suggested. As shown in Table 6, both high convergent validity and discriminant validity were suggested.

To check the validity of second-order formative constructs, the weights of first-order constructs were checked following Cenfetelli and Bassellier [81]’s steps. As shown in Table 7, for the two second-order formative constructs, i.e., perceived design esthetics and smartphone symbolic use, the weights of their first order constructs were all significant.

### 5.2. Structural Model

#### 5.2.1. Direct Effect

Partial least squares results for the direct hypotheses of the research model are presented in Figure 4. H1a and H1b, about the positive relationship between the two enriching factors (self-expressive symbolism and categorical symbolism) and mobile identity, are supported.

H2 concerns the positive effect of perceived design esthetics as the gratifying factor on mobile identity via the mediation of the two enriching factors. We tested the mediating effect following a three-step procedure [82]: (1) there should be a significant relationship between IV and DV; (2) there should be a significant relationship between IV and M (mediator); (3) the regression effect of DV on M and IV should be tested at the same time. A partial mediation effect can be established if the difference between IV and DV values in step 3 is smaller than in step 1; while a full mediation effect can be established if the relationship between IV and DV becomes insignificant in step 3. According to the results shown in Table 8, the difference between values of IV (perceived design esthetics) and DV (mobile identity) were smaller in step 3, so a partial mediating effect of the two enriching factors is confirmed, providing support to H2. Finally, H3 about the positive relationship between mobile identity and smartphone symbolic use (β = 0.270, t = 4.008) was supported.

The control variable “time since ownership” tries to tease out the intervening effect of time. It was shown to negatively influence symbolic usage (β = −0.090, t = 2.170), suggesting that as time goes by, users’ non-functional usage behavior wanes. Additionally, brand attachment was positively related with symbolic use (β = 0.319, t = 5.055).

In summary, the results support all the hypotheses about the direct effects in the research model: 26.7% variance of smartphone symbolic use was explained by mobile identity together with control variables. The enriching factors and gratifying factor altogether explained 35.1% variance of mobile identity.

#### 5.2.2. Moderating Effect

To test the moderating effect of individual materialism, the interaction variable was set as the product of the two interacting variables following the procedures suggested by [83]. Then, hierarchical analysis was conducted.

As shown in Table 9, in model 1, the three antecedents of mobile identity were included, as well as the moderator. In model 2, the interaction term between IM (individual materialism) and self-expressive symbolism was added to the model. The R-square changed from 36.6% to 37.7% (f^2^ = 0.017), which is lower than the 0.02 threshold. The path coefficient also suggested a non-significant interaction effect (β = 0.108, t = 1.108). H4a is thus not supported. In model 3, the interaction term between IM and categorical symbolism was added to the model. The results show model 3 had an elevated R-square from 0.366 to 0.380 (f^2^ = 0.026), suggesting a small effect size. The results also show that the path coefficient of the interaction term was positive and significant (β = 0.121, t = 2.012), supporting H4b. The interaction effect is delineated in Figure 5. As shown in the figure, the effect of categorical symbolism of mobile identity is stronger under high IM, confirming the positive moderating effect.

## 6. Discussion

### 6.1. Summary of Key Findings

This paper conceptualized non-functional smartphone use as symbolic use, and attempted to understand such behavior using attachment theory. Mobile identity was identified as an important antecedent of symbolic use. Mobile identity in turn was formed by mobile symbolism and mobile design esthetics. A contingent effect of personal characteristics was found. Several interesting findings are derived from the current study.

First, this paper shows that attachment theory is a proper theory in explaining smartphone symbolic use. The special nature of smartphones as a personal extension and identity symbol with a close bond is well captured in this theory. Mobile identity, as a symbolic and emotional aspect of mobile attachment, was confirmed as an important predictor of smartphone symbolic use.

Second, the enriching factor, perceived mobile symbolism with two subdimensions (categorical symbolism and self-expressive symbolism), was proved to be an important antecedent of mobile identity. The enriching feature of smartphones, i.e., that they act as surrogates of users’ identity expression, was largely ignored in previous studies. Based on theory building and hypothesis testing, we conclude that smartphones’ symbolic properties and non-functional smartphone usage behavior are especially pertinent owing to their unique role as users’ self-extension to express personal identity.

Third, the significance of the gratifying factor, i.e., design esthetics, should be particularly emphasized in the smartphone context according to the findings of the current paper. It proved that design esthetics can exert an effect on usage behavior through the mechanism of symbolic association. In other words, beautifully designed smartphones (both in terms of its software and hardware design) lead to the establishment of the symbolic bond between smartphones and users, which in turn influences symbolic usage of smartphones. While most mobile usage literature underestimates the role of design esthetics in affecting smartphone usage, the results remind us that we should never downplay the gratifying function of smartphones, as it is a fundamental part of the user experience. This is especially the case in the current research context because smartphones are carried around by users all the time, and are constantly used and exposed in public, therefore they become users’ special personal extensions.

Last but not least, some interesting findings about the moderating role of individual characteristics in the process of attachment formation were also revealed. Although individual materialism is hypothesized to moderate both the effect of categorical symbolism and self-expressive symbolism, the results only show the positive moderating role of individual materialism in the relationship between categorical symbolism and mobile identity. That is to say, when forming a symbolic connection with mobile technologies, users who rate highly in individual materialism focus more on whether the mobile technology can help them express their ideal image but not actual image. A possible explanation is that people who have high individual materialism are highly concerned about how possessions can help them promote their image [85]. Compared to self-expressive symbolism, in which objects serve as mirrors of actual identity, the categorical symbolism of mobile technology can better serve the purpose of promoting self-image by presenting an ideal identity.

### 6.2. Implications

Broadly speaking, this paper is among the first to conceptualize and understand non-functional smartphone usage. By studying smartphone symbolic use, it enriches the mobile usage literature by illustrating the uniqueness of the smartphone usage context. Based on the close bond between users and smartphones, the attachment theory from marketing literature was introduced to understand smartphone symbolic use. The mobile attachment antecedents, i.e., gratifying and enriching factors specific to the mobile context, were identified. The paper also contributes to the original attachment theory by integrating it with the self-concordance model to understand the contingent factors in the attachment formation process. We present the specific theoretical and practical contributions as follows.

First, the paper delineates the unique characteristics of the smartphone usage context. Specifically, the uniqueness is reflected by the deep relationships users have with smartphones and their role as a combination of both a pragmatic tool and symbolic object. Based on such distinct characteristics, the paper re-conceptualized the technology usage construct. Given the importance of the usage construct, it is crucial to capture the uniqueness of the usage context in conceptualization [86]. While most previous mobile usage studies adopt a lean and unified conceptualization of usage based on factors such as duration and frequency [13,14,15], this paper is among the first to contextualize the mobile usage construct. Specifically, we bring to light the symbolic property of smartphones. One new type of usage behavior, symbolic use, is proposed. It systematically synthesizes the scatter research about non-functional usage behavior in the previous literature [9,20,21] and has strong economic implications. Among the three types of symbolic use (i.e., mobile personalization, public display, and protective behavior), mobile personalization is of highest interest to practitioners as personalization entails purchase of mobile accessories. The profit of this industry is substantial. Realizing its importance, this paper identified mobile identity as a key antecedent of symbolic usage. The formation of mobile identity is also discussed by enhancing design esthetics and perceived mobile symbolism. Mobile accessory makers can thus learn important lessons from this study about how to attract more customers.

Second, this paper theorized on and proved the significance of a few smartphone-specific antecedents that have rarely caught researchers’ attention. To begin with, we bring to light role of design esthetics in the study. Its working mechanism, i.e., symbolic association [60], is introduced and confirmed. Equally importantly, mobile symbolism, as a key characteristic that distinguishes smartphones from other personal technologies, enriches users’ self-concept by allowing them to express their actual self-identity (through self-expressive symbolism) or ideal self-identity (through categorical symbolism). Such symbolic meaning of mobile devices is very important for the formation of mobile identity and therefore leads to identity-related usage of mobile devices. The symbolic aspect of smartphones calls for practitioners’ attention. To successfully form users’ mobile identity, both self-expressive symbolism and categorical symbolism should be emphasized. Practitioners can cultivate smartphone symbolism through both product design and marketing. When designing their product, producers should understand the identity of their target consumers and match the consumers’ characteristics in the product design, or provide more options to consumers by allowing them to customize their own devices to match their personal identity. Moreover, the marketing campaigns of smartphones should not only emphasize the functional advancement of the product, but also advocate the symbolic meaning of the product. For instance, in advertisements, marketers can create a categorical or ideal social image for their potential customers, e.g., that stylish, cool, or successful users like to use their products. Associating products with socially desirable characters can facilitate the formation of mobile symbolism.

Finally, the current paper also contributes back to the attachment theory by integrating it with the goal-concordance model. While the original attachment only concerns the direct relationship between need satisfaction and attachment formation, it has not considered the potential contingent factors, i.e., whether such a process differs in different circumstances. The goal-concordance model addresses this gap by arguing that the need satisfaction process leads to positive outcomes only when the need satisfied is consistent with users’ interests and values. By integrating the goal-concordance model with the attachment model, the moderator of individual materialism was confirmed to intervene in the attachment formation process. This also answers questions from the previous debate about the positive or negative influence of smartphone usage [54,55,56,57] by proving that the consequence of goal attainment relies on whether it is consistent with individual characteristics.

### 6.3. Limitations

To begin with, the methodology used in this dissertation was a self-reported survey. Although the method suited the research purpose well and boasts several advantages, it also suffers several disadvantages such as self-selection bias and common method bias (CMB). Although data analysis in this study showed that CMB was not a major concern for this study, some other methods or measurements, such as an objective daily usage log, should be considered in future study to further confirm the results.

Second, this was a cross-sectional study that measured both independent variables and dependent variables at the same point of time. A longitudinal study that can capture the process of attachment formation was not performed at the current stage. This may potentially compromise the internal validity of the study. However, it is hoped this concern is partially alleviated by the sound theoretical background. It would be interesting for future research to test this model using a longitudinal design. Such a design could better reveal how the attachment to mobile devices is gradually formed as users’ experience with certain smartphone increases. It would also be interesting to know the difference between users’ attachment style and behavioral style in different stages of usage.

Meanwhile, although smartphone usage behavior is a global issue, all the participants of the current study were from mainland China due to the sampling method. As a result, cultural influence was not included in the current paper. Cultural issues can potentially influence users’ interests and values, and therefore act as moderators intervening in attachment formation. Future studies may consider such topics.

Finally, the construct of smartphone symbolic use is worth further looking into. Future study could test each of the three subdimensions in greater detail. For example, they could be broken into three individual constructs, then future research could explore whether there was any difference in their forming mechanism.

## 7. Conclusions

Due to smartphones’ personal nature and high visibility in use, they are often regarded as personal extensions and used for identity expression. Based on the distinct characteristics of the smartphone usage context, this study is among the first to conceptualize and study smartphone symbolic use. Attachment theory was introduced to the smartphone context to identify the motive behind such behavior. Mobile identity was confirmed as an important antecedent of symbolic use, and the formation of mobile identity can be facilitated by mobile symbolism and design esthetics. Based on the self-concordance model, individual materialism is found to moderate the attachment formation process.

The current paper enriches the mobile usage literature by illustrating the uniqueness of the smartphone context and contextualizing the smartphone usage construct. This paper theorized on and proved the significance of a few smartphone-specific antecedents such as design esthetics and mobile symbolism. It also contributes to attachment theory by integrating it with the goal-concordance model.

## Figures and Tables

**Figure 1 ijerph-19-14036-f001:**
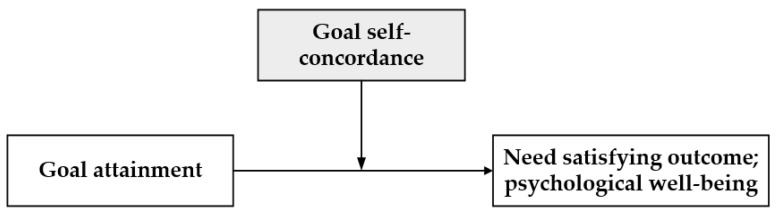
Self-concordance model.

**Figure 2 ijerph-19-14036-f002:**
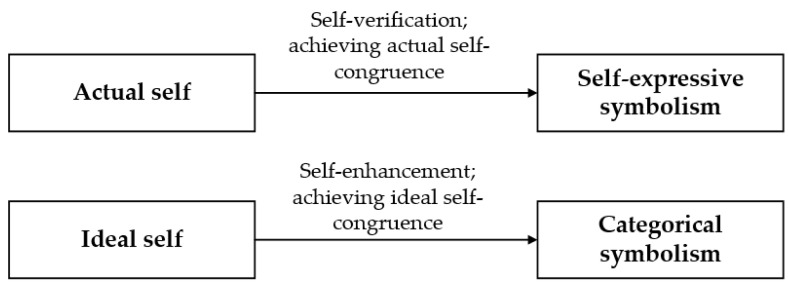
Working mechanism of self-expressive and categorical symbolism.

**Figure 3 ijerph-19-14036-f003:**
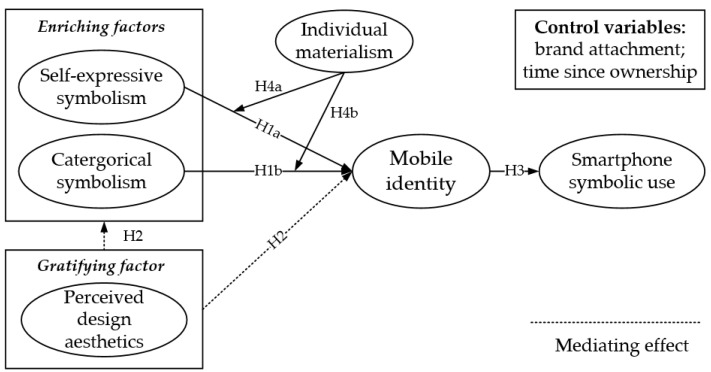
Research model of the effects of mobile identity on smartphone symbolic use.

**Figure 4 ijerph-19-14036-f004:**
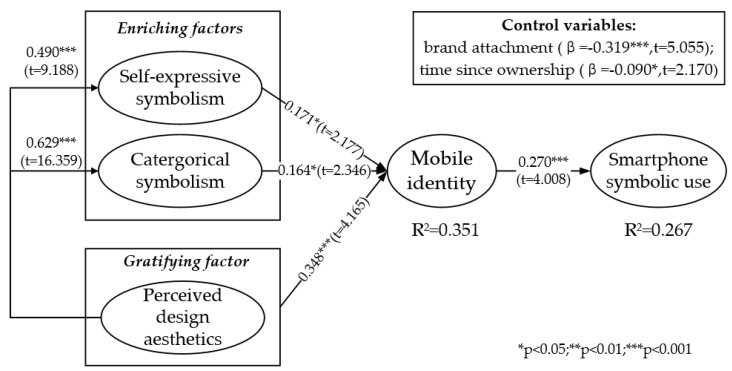
Structural model.

**Figure 5 ijerph-19-14036-f005:**
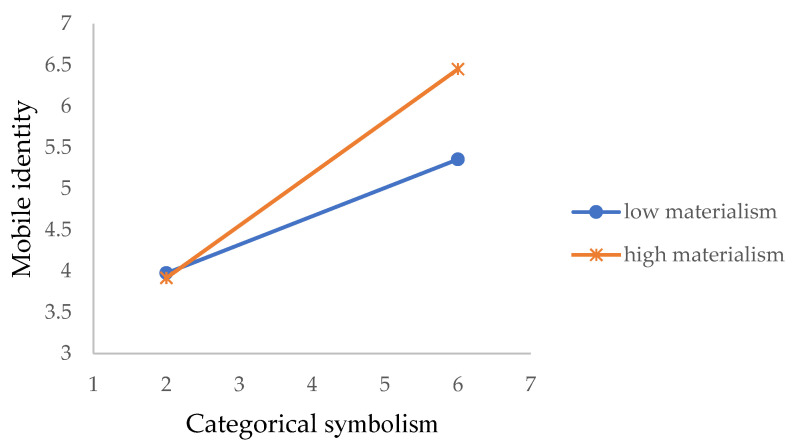
Interaction effect between categorical symbolism and individual materialism.

**Table 1 ijerph-19-14036-t001:** Definition of smartphone symbolic use and its underlying dimensions.

Definition of Smartphone Symbolic Use	Underlying Dimensions	Definition	Source
Users’ employment of mobile technology for non-functional purposes and social outcomes, which are mostly related to self-identity expression.	Mobile personalization	The extent to which users customize mobile appearance to express themselves through both interior and exterior decoration.	[10,16]
	Public display	The extent to which a user likes to display a smartphone in public even with no specific intention of using its functions.	[11,23,26]
	Protective behavior	The extent to which a user uses a smartphone with care and worry about its deterioration.	[24,25]

**Table 2 ijerph-19-14036-t002:** Demographic Information.

Variable	Category	Frequency	Percentage
**Gender**	Male	130	47.9%
	Female	141	52.1%
	Total	271	100%
**Age**	≤20	46	17.0%
	21–30	210	77.5%
	31–40	14	5.1%
	≥41	1	0.4%
	Total	271	100%
**Smartphone brand**	Apple	84	31.0%
	Huawei	58	21.4%
	Xiaomi	27	10%
	Samsung	23	8.5%
	Others	79	29.1%
	Total	271	100%

**Table 3 ijerph-19-14036-t003:** Measurement items.

Construct	Measurement Items	Source
**Self-expressive symbolism**	Respondents were given the following instructions: Take a moment to think about your smartphone. Describe this device using personality characteristics such as fashionable (or outdated), high-tech (or low-tech), innovative (or conservative), etc. Now think about how you see yourself. What kind of person are you? How would you describe your personality? Once you’ve done this, indicate your agreement or disagreement with the following statements: **Self-expressive symbolism (SES.)** **SES1**: The personality of my smartphone is consistent with how I see myself (my actual self). **SES2**: The personality of my smartphone is a mirror image of me (my actual self).	Adapted from [39]
**Categorical symbolism**	Respondents were given the following instructions: Take a moment to think about your smartphone. Describe this device using personality characteristics such as fashionable (or outdated), high-tech (or low-tech), innovative (or conservative), etc. Now think about how you would like to see yourself (your ideal self). What kind of person would you like to be? Once you’ve done this, indicate your agreement or disagreement with the following statements: **Categorical symbolism (CS.)** **CS1**: The personality of my smartphone is consistent with how I would like to be (my ideal self). **CS2**: The personality of my smartphone is a mirror image of the personal I would like to be (my ideal self).	Adapted from [39]
**Perceived design esthetics**	**Perceived hardware design esthetics (HA.)** **HA1**: I find the design of my smartphone looks pleasant. **HA2**: I think the design of my smartphone is esthetical. **HA3**: I think the design of my mobile device is fascinating. **HA4**: I find the design of my mobile device to be creative.	Adapted from [63,65]
	**Perceived software design esthetics (SA.)** **SA1**: I find the software design in my smartphone looks pleasant. **SA2**: I think the software design in my smartphone is esthetical. **SA3**: I think the layout of software in my smartphone is fascinating. **SA4**: I find the software design in my smartphone to be creative.	
**Mobile identity**	**Mobile identity (MI.)** **MI1**: I feel my smartphone is a part of me. **MI2**: My smartphone is very special to me. **MI3**: I am very attached to my smartphone. **MI4**: I identify strongly with my smartphone.	Adapted from [62]
**Smartphone symbolic use**	**Public display (PD.)** **PD1**: I like to display my smartphone to others in public even when I do not need to use it. **PD2**: I like to put my smartphone in public places so that others can see it. **PD3**: I like others in the public to see my smartphone. **PD4**: I like to show my smartphone to others.	Self-developed
	**Mobile personalization (PL.)** **PL1**: I like to customize the appearance of my smartphone a lot. **PL2**: I like to change the physical appearance of my smartphone by purchasing some decorative accessories (e.g., protective case). **PL3**: I have bought many decorative accessories for my smartphone.	Self-developed based on descriptions of [10,16]
	**Protective behavior (PB.)** **PB1**: I spend great effort in protecting my smartphone. **PB2**: I use my smartphone with care. **PB3**: If my smartphone deteriorates, I will be sad.	Adapted from [24,25]
**Individual materialism**	**Individual Materialism (IM.)** **IM1**: I admire people who own expensive homes, cars, and clothes. **IM2**: Buying things gives me a lot of pleasure. **IM3**: I like a lot of luxury in my life. **IM4**: I’d be happier if I could afford to buy more things.	Adapted from [75]

**Table 4 ijerph-19-14036-t004:** Descriptive statistics and reliability.

Variable	Item No.	Mean	S.D.	Alpha	C.R.	AVE
**SES**	2	4.456	1.295	0.919	0.962	0.925
**CS**	2	4.563	1.395	0.903	0.954	0.911
**HA**	4	4.896	1.258	0.928	0.949	0.824
**SA**	4	4.761	1.162	0.931	0.951	0.829
**MI**	4	4.804	1.189	0.889	0.922	0.749
**PD**	4	2.771	1.263	0.938	0.956	0.844
**PL**	3	3.723	1.493	0.846	0.901	0.761
**PB**	3	5.235	1.227	0.849	0.908	0.767
**IM**	4	4.680	1.041	0.761	0.847	0.582

Note: (1) Alpha—Cronbach’s alpha; C.R.—composite reliability; AVE—average variable extraction; (2) SES—self-expressive symbolism; CS—categorical symbolism; HA—hardware esthetics; SA—software esthetics; MI—mobile identity; PD—public display; PL—mobile personalization; PB—protective behavior; IM—individual materialism.

**Table 5 ijerph-19-14036-t005:** Discriminant validity.

	SES	CS	HA	SA	MI	PD	PL	PB	IM
**SES**	**0.962**								
**CS**	0.672	**0.955**							
**HA**	0.432	0.589	**0.908**						
**SA**	0.448	0.550	0.618	**0.911**					
**MI**	0.455	0.503	0.436	0.523	**0.865**				
**PD**	0.280	0.342	0.337	0.335	0.261	**0.919**			
**PL**	0.191	0.188	0.306	0.242	0.219	0.290	**0.873**		
**PB**	0.194	0.274	0.362	0.344	0.364	0.213	0.311	**0.876**	
**IM**	0.152	0.226	0.170	0.130	0.232	0.084	0.197	0.237	**0.763**

Note: constructs’ square root of AVE is shown in black background.

**Table 6 ijerph-19-14036-t006:** Loadings and cross-loadings.

	SES	CS	HA	SA	MI	PD	PL	PB	IM
**SES1**	0.96	0.66	0.41	0.41	0.43	0.24	0.17	0.21	0.14
**SES2**	0.96	0.63	0.42	0.45	0.44	0.30	0.20	0.17	0.18
**CS1**	0.63	0.95	0.54	0.50	0.43	0.33	0.18	0.28	0.25
**CS2**	0.66	0.96	0.58	0.54	0.52	0.33	0.18	0.25	0.25
**HA1**	0.36	0.55	0.86	0.59	0.38	0.24	0.25	0.37	0.20
**HA2**	0.40	0.55	0.95	0.57	0.40	0.29	0.28	0.37	0.17
**HA3**	0.40	0.53	0.94	0.55	0.41	0.38	0.30	0.29	0.16
**HA4**	0.40	0.51	0.88	0.54	0.39	0.30	0.29	0.28	0.15
**SA1**	0.34	0.45	0.47	0.88	0.42	0.21	0.17	0.34	0.20
**SA2**	0.44	0.52	0.61	0.94	0.48	0.35	0.22	0.32	0.14
**SA3**	0.42	0.49	0.57	0.93	0.54	0.34	0.23	0.29	0.14
**SA4**	0.43	0.53	0.60	0.89	0.46	0.31	0.27	0.31	0.14
**MI1**	0.30	0.32	0.26	0.38	0.83	0.15	0.13	0.29	0.25
**MI2**	0.42	0.46	0.38	0.48	0.88	0.23	0.20	0.31	0.19
**MI3**	0.36	0.41	0.35	0.48	0.90	0.25	0.15	0.24	0.22
**MI4**	0.47	0.51	0.48	0.46	0.85	0.26	0.25	0.40	0.20
**PD1**	0.24	0.36	0.38	0.34	0.25	0.89	0.29	0.21	0.18
**PD2**	0.26	0.31	0.27	0.26	0.23	0.93	0.25	0.17	0.16
**PD3**	0.26	0.27	0.27	0.30	0.21	0.93	0.23	0.17	0.16
**PD4**	0.27	0.31	0.31	0.32	0.26	0.93	0.29	0.22	0.18
**PL1**	0.19	0.21	0.30	0.19	0.15	0.20	0.82	0.27	0.18
**PL2**	0.14	0.14	0.25	0.21	0.19	0.24	0.91	0.31	0.11
**PL3**	0.18	0.16	0.26	0.23	0.22	0.30	0.89	0.24	0.11
**PB1**	0.22	0.23	0.29	0.30	0.26	0.17	0.28	0.87	0.12
**PB2**	0.17	0.24	0.32	0.29	0.32	0.16	0.24	0.91	0.14
**PB3**	0.13	0.25	0.34	0.31	0.35	0.22	0.29	0.85	0.19
**IM1**	0.12	0.14	0.12	0.18	0.16	0.19	0.00	0.06	0.71
**IM2**	0.19	0.26	0.19	0.14	0.21	0.09	0.24	0.23	0.80
**IM3**	0.12	0.26	0.14	0.12	0.18	0.25	0.08	0.04	0.78
**IM4**	0.07	0.13	0.10	0.08	0.19	0.06	0.10	0.19	0.75

Note: items loadings on their corresponding constructs are shown in black background.

**Table 7 ijerph-19-14036-t007:** Weights of formative constructs.

Second-Order Construct	First-Order Construct	Weights	T-Statistics
**Perceived design esthetics**	Perceived hardware design esthetics	0.320	2.420
	Perceived software design esthetics	0.771	7.198
**Smartphone symbolic use**	Public display	0.403	3.609
	Mobile personalization	0.260	2.293
	Protective behavior	0.680	7.344

**Table 8 ijerph-19-14036-t008:** Three-step test of mediating effect.

IV	M	DV	IV→DV	IV→M	M→DV	IV→DV	Result
Perceived design esthetics	Self-expressive symbolism	Mobile identity	0.541 (t = 11.905)	0.490 (t = 8.174)	0.457 (t = 9.835)	0.419 (t = 6.079)	Partial
Perceived design esthetics	Categorical symbolism	Mobile identity	0.541 (t = 11.905)	0.629 (t = 16.517)	0.506 (t = 12.470)	0.366 (t = 4.572)	Partial

**Table 9 ijerph-19-14036-t009:** Moderating effect of individual materialism.

Contingent Effects (DV = Mobile Identity)			
	Model 1	Model 2	Model 3
Perceived design esthetics (PDA)	0.350 *** (t = 4.441)	0.345 *** (t = 4.742)	0.328 *** (t = 4.233)
Self-expressive symbolism (SES)	0.167 * (t = 2.159)	0.156 * (t = 1.973)	0.172 * (t = 2.542)
Categorical symbolism (CS)	0.146 * (t = 1.982)	0.148 * (t = 1.996)	0.144 + (t = 1.808)
Individual materialism (IM)	0.117 * (t = 2.051)	0.128 * (t = 2.076)	0.135 * (t = 2.431)
IM * SES		0.108 n.s. (t = 1.108)	
IM * CS			0.121 * (t = 2.012)
R^2^	0.366	0.377	0.380
Adj. R^2^	0.356	0.365	0.368
R^2^ Change		0.011	0.014
f^2^-statistics		0.017	0.026

Note: Cohen’s f^2^ is one of several effect size measures for multiple regression, and the f^2^ effect size measure for hierarchical multiple regression is: f^2^ = [R^2^AB − R^2^A]/[1 − R^2^AB] [84]. R^2^A is the variance accounted for by a set of one or more independent variables A, and R^2^AB is the combined variance accounted for by A and another set of one or more independent variables B. f^2^ of 0.02, 0.15, 0.30 are termed small, medium, and large effect sizes, respectively. + *p* < 0.1, * *p* < 0.05, ** *p* < 0.01, *** *p* < 0.001, n.s. = not significant.

## Data Availability

Data of this study can be accessed upon personal request.

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
