# Peer review of "Effects of Mobile Identity on Smartphone Symbolic Use: An Attachment Theory Perspective"

_ijerph, 2022, doi:10.3390/ijerph192114036_

Round 1
Reviewer 1 Report
Please see attached file

Author Response
Thank you for your great comments for making this paper better. All the comments have been replied to and carefully addressed in this version. Please refer to the attached document for details.

Reviewer 2 Report
Thank you for the invitation to review this article. The topic of the paper is very interesting. Find the feedback for the paper below:
1. I recommend to the authors to better contextualize the discussion of their study in the field of health This is an important aspect in order for the text to be suitable to the IJERPH journal.
2. We recommend that authors accurately cite the references of the definitions used in their study. For example, Mobile personalization is defined without specifying the source. This also occurs with other concepts used. See also Table 1.
3. Figure 1. Research model. - the authors should chose a title for the proposed research model.
4. Taking into account the small size of their sample, the authors should argue with sources from the methodological literature that the number of valid responses allows the type of analysis performed.
5. For the measurement model validity, the authors should report also if the model is reflective or formative (the measurement theory) and what sources from the literature can be used as a references for this conceptualization. Also, the Discriminant Validity section of the results report should include also the Heterotrait-Monotrait Ratio (HTMT). The authors should also report the factor loadings. The data interpretation needs to be based on methodological references. Overall, a more rigorous approach is needed in this section.
6. The authors should title the Figure 2. Also, the data interpretation of the structural model needs to be based on methodological references.
7. Research discussion should be extended and correlated with existing studies from the literature.
8. The limitation section of the study should also be extended. Future research studies should be clearly presented based on this work.
Author Response
Thank you for your great comments for making this paper better! All the comments have been replied to and addressed in this version, please refer to the attached document for greater detail. Thank you again!

Round 2
Reviewer 1 Report
Thanks for the reply to the comments and the edits applied to the manuscript. No further comments or requests on my side.
All the best for your future work!